# Depressive Symptoms during Pregnancy: Prevalence and Correlates with Affective Temperaments and Psychosocial Factors

**DOI:** 10.3390/jpm13020211

**Published:** 2023-01-25

**Authors:** Marianna Mazza, Carla Avallone, Georgios Demetrios Kotzalidis, Giuseppe Marano, Lorenzo Moccia, Anna Maria Serio, Marta Balocchi, Ilenia Sessa, Delfina Janiri, Ilaria De Luca, Caterina Brisi, Maria Chiara Spera, Laura Monti, Angela Gonsalez del Castillo, Gloria Angeletti, Daniela Chieffo, Lucio Rinaldi, Luigi Janiri, Antonio Lanzone, Giovanni Scambia, Eugenio Maria Mercuri, Gabriele Sani

**Affiliations:** 1Institute of Psychiatry and Psychology, Department of Geriatrics, Neuroscience and Orthopedics, Fondazione Policlinico Universitario A. Gemelli IRCCS, Università Cattolica del Sacro Cuore, 00168 Rome, Italy; 2NESMOS Department (Neurosciences, Mental Health, and Sensory Organs), Faculty of Medicine and Psychology, Sapienza University of Rome, 00189 Rome, Italy; 3Unit of Clinical Psychology, Fondazione Policlinico Universitario Agostino Gemelli IRCCS, Università Cattolica del Sacro Cuore, 00168 Rome, Italy; 4Department of Obstetrics and Gynaecology, Fondazione Policlinico Universitario Agostino Gemelli IRCCS, Università Cattolica del Sacro Cuore, 00168 Rome, Italy; 5Division of Gynecologic Oncology, Department of Woman, Child and Public Health, Fondazione Policlinico Universitario A. Gemelli IRCCS, Università Cattolica del Sacro Cuore, 00168 Rome, Italy; 6Paediatric Neurology Unit, Fondazione Policlinico Agostino Gemelli IRCCS, Università Cattolica del Sacro Cuore, 00168 Rome, Italy

**Keywords:** women, affective temperaments, pregnancy, peripartum, depression, psychosocial factors

## Abstract

Pregnancy is a unique experience in women’s life, requiring a great ability of adaptation and self-reorganization; vulnerable women may be at increased risk of developing depressive symptoms. This study aimed to examine the incidence of depressive symptomatology during pregnancy and to evaluate the role of affective temperament traits and psychosocial risk factors in predicting them. We recruited 193 pregnant women, collected data regarding sociodemographic, family and personal clinical variables, social support and stressful life events and administered the Mood Disorder Questionnaire (MDQ), the Patient Health Questionnaire-9 (PHQ-9), and the Temperament Evaluation of Memphis, Pisa, Paris and San Diego-Autoquestionnaire (TEMPS-A). In our sample, prevalence of depressive symptomatology was 41.45% and prevalence of depression was 9.85% (6.75% mild and 3.10% moderate depression). We have chosen a cutoff >4 on PHQ-9 to identify mild depressive symptoms which may predict subsequent depression. Statistically significant differences between the two groups were found in the following factors: gestational age, occupation, partner, medical conditions, psychiatric disorders, family psychiatric history, stressful life events, and TEMPS-A mean scores. In our sample mean scores on all affective temperaments but the hyperthymic, were significantly lower in the control group. Only depressive and hyperthymic temperaments were found to be, respectively, risk and protective factors for depressive symptomatology. The current study confirms the high prevalence and complex aetiology of depressive symptomatology during pregnancy and suggests that affective temperament assessment seems to be a useful adjunctive instrument to predict depressive symptomatology during pregnancy and postpartum.

## 1. Introduction

Pregnancy is a unique experience in women’s life, a complex bio-psycho-social process, characterized by anatomical, physiological, biological, psychological, and social changes. This critical period requires a great ability of adaptation and self-reorganization; vulnerable women may be at increased risk of developing depressive symptoms. The prevalence of prenatal depression is estimated to range from 7% to 20% in high income countries [1]. Results of an umbrella review show high prevalence of antenatal depression, especially in low income countries, ranging from 15% to 65% [2]. Depression during pregnancy has been linked to hypertension, preeclampsia, gestational diabetes, preterm birth, low birth weight, foetal growth restriction, postnatal complications as well as negative impact on child development [2,3].

In a systematic review, maternal anxiety, life stress, history of depression, lack of social support, unintended pregnancy, Medicaid insurance, domestic violence, lower income, lower education, smoking, single status, and poor relationship quality showed associations with antenatal depressive symptoms [4]. A recent review identified three main groups of antenatal depressive risk factors: sociodemographic, obstetric, and psychological ones. Among these variables, the main ones were low level of education and income, unplanned pregnancy, history of psychological disorders, depression, anxiety, and low social support during pregnancy [5].

Temperament refers to the temporally stable biological ‘core’ of personality and may be seen as a bridge between the psychology and the biology of affective disorders [6]. Akiskal and colleagues developed a five affective temperaments model (depressive, cyclothymic, hyperthymic, irritable, and anxious) and the TEMPS-A, a psychometric instrument to assess them [7]. Research shows that up to 20% of the population has some kind of marked affective temperaments which may be considered subclinical forms and antecedents of mood disorders [6]. A meta-analysis, evaluating affective temperaments, suggests a continuum model of affective temperament domains spanning from healthy controls to mood disorders [8]. Affective temperaments could also play a pathoplastic role influencing the emergence and clinical evolution of affective disorders and their characteristics: predominant polarity, symptomatic expression, long-term course and consequences, response and adherence to treatment and outcome as well [9].

To the best of our knowledge, few studies have investigated affective temperaments during pregnancy. The present study aimed to examine the incidence of depressive symptomatology during pregnancy and to investigate the role of affective temperament traits and other possible psychosocial risk factors in predicting depression.

## 2. Materials and Methods

This cross-sectional study was conducted from July 2020 to November 2021. We consecutively recruited 200 pregnant women at the Gynaecology and Obstetrics Day Hospital of the Fondazione Policlinico Universitario A. Gemelli IRCCS. Exclusion criteria were age less than 18 years old, failure to provide free informed consent, diagnosis of psychosis, and incomplete comprehension of the Italian language that prevented women from completing the study protocol. Overall, of the 200 eligible women, 193 participated in the study; five refused to sign the consent form and two were not sufficiently fluent in Italian. At assessment we performed the completion of a sociodemographic data collection form, including family and personal clinical variables, and the administration of specific questionnaires. We also collected data regarding social support and stressful life events by posing specific questions as delineated in Paykel et al. [10]. Specifically, women were asked about conflicts with their family of origin, about their marital adjustment, and about past events that might have affected their pregnancy (i.e., loss of a significant other, life-threatening disease or event in oneself or significant other, professional stress, and economic hardship) and the timing and perceived intensity of each event was recorded by the interviewing clinician. Eventually, each woman was classified as having or not having social support and as having or not having a history of stressful life events. We also administered the Mood Disorder Questionnaire, the Patient Health Questionnaire-9, and the Temperament Evaluation of Memphis, Pisa, Paris and San Diego-Autoquestionnaire.

The Mood Disorder Questionnaire (MDQ) is a questionnaire developed by Hirschfeld et al. as screening instrument for bipolar disorders. It is based on DSM-IV criteria and consists of 3 “yes/no” questions. The first one examines the lifelong history of manic or hypomanic symptoms and consists of 13 items. The second question explores the occurrence of symptoms during the same period and the third one how significantly they impacted on daily functioning. Positive screen requires 7 or more symptoms, multiple symptoms occurring at the same time, and notable psychosocial impairment [11,12]. We used the validated Italian version of the MDQ [13].

The Patient Health Questionnaire-9 (PHQ-9) is based on DSM-IV criteria to assess the presence of depressive symptoms in the preceding 2 weeks. It consists of 9 questions scored from 0 (not at all) to 3 (nearly every day), for a total score ranging from 0 to 27. A score of 0–4 is considered as having minimal depressive symptoms, 5–9 mild depressive symptoms, 10–14 mild depression, 15–19 moderate depression, 20–27 severe depression [14,15]. We used the validated Italian version of the PHQ-9 [16]. Based on the results obtained on this instrument, we subdivided the sample into two subgroups; women scoring 4 or less entered the PHQ-9 ≤4 group, while those scoring >4 were considered the symptomatic group, independently from the severity of their depression.

The Temperament Evaluation of Memphis, Pisa, Paris and San Diego-Autoquestionnaire (TEMPS-A) was used to assess the five affective temperaments in psychiatric and healthy subjects: depressive (D), cyclothymic (C), hyperthymic (H), irritable (I), and anxious (A). It is an autoquestionnaire that contains 110 “true” or “false” items. The highest score is considered to indicate the prevailing temperament [7]. We used the validated Italian version of TEMPS-A [17]. The research protocol was approved by the Ethical Committee of Fondazione Policlinico Agostino Gemelli IRCCS, Università Cattolica del Sacro Cuore, Rome, Italy (Protocol ID:2221).

## 3. Statistical Analysis

We subdivided our sample into two groups according to the PHQ-9 cut-off score >4, i.e., a PHQ-9 ≤4 and a PHQ >4 group. Analyses used standard comparisons of continuous measures (ANOVA) and categorical measures (contingency table/*χ*^2^) to compare factors of interest in the two groups. Factors significantly associated with depressive symptomatology underwent a binomial logistic regression to generate Odds Ratios (ORs) and their 95% Confidence Intervals (CIs). We examined possible collinearity between variables of interest by ensuring that the variance inflation factor (VIF) indicator obtained from linear regression analysis was <3.5. We used the statistical routines of the SPSS (Statistical Package for the Social Sciences) 24.0 for Windows software (IBM Co., Armonk, New York, NY, USA, March 2016).

## 4. Results

In our sample (*n* = 193), nobody reached the severe depression cut-off score on the MDQ and were divided into two groups according to their scores on the PHQ-9: 113 (58.55%) scored 4 or lower (mean = 2.32 ± 1.33), 80 (41.45%) higher than 4 (mean = 8.20 ± 3.20) (Figure 1). Thus, the >4 on the PHQ-9 group comprised women with mild depressive symptoms, mild depression, and moderate depression. Prevalence of depressive symptomatology was distributed as follows: mild depressive symptoms *n* = 61 (31.60%), mild depression *n* = 13 (6.75%) and moderate depression *n* = 6 (3.10%). Nobody scored 20 or higher.

Sociodemographic variables, epidemiologic characteristics and temperament scores are shown in Table 1.

Statistically significant differences between the two groups were found in the following factors: gestational age (*χ*^2^ = 6.32; *p* = 0.042), occupation (*χ*^2^ = 4.92; *p* = 0.027), partner (*χ*^2^ = 4.48; *p* = 0.034), medical conditions (*χ*^2^ = 4.17; *p* = 0.041), psychiatric disorders (*χ*^2^ = 11.5; *p* < 0.001), family psychiatric history (*χ*^2^ = 5.71; *p* = 0.017), stressful life events (*χ*^2^ = 5.11; *p* = 0.024), TEMPS-A depressive (F = 42.20; *p* < 0.001), TEMPS-A cyclothymic (F = 16.36; *p* < 0.001), TEMPS-A hyperthymic (F = 16.49; *p* < 0.001), TEMPS-A irritable (F = 21.66; *p* < 0.001), and TEMPS-A anxious (F = 38.60; *p* < 0.001).

In the PHQ-9 ≤4 group, mean age was 35.39 ± 5.50 years, ranging 20–45 years. In this sample, there were 107 Italian citizens (94.70%) and 6 foreigners (5.30%). In the PHQ-9 >4 group mean age was 33.77 ± 6.07 years, ranging 19–45 years. There were 73 Italian citizens (91.25%) and 7 foreigners (8.75%). Results for sociodemographic and clinical variables are shown in Table 1. Briefly, the two groups did not differ for age, although the PHQ-9 >4 tended to be nominally older. The distribution per trimester showed the PHQ-9 >4 group to have more pregnant women in the first trimester, but this finding is not strong (*p* = 0.04). Having a job and a partner was more frequent in the PHQ-9 ≤4 group, while having a past medical condition, a lifetime psychiatric disorder, a family psychiatric history, and past stressful life events was more frequent in the PHQ-9 >4 group. Stressful life events were present in 52 (46%) of women in the PHQ-9 ≤4 group [job loss *n* = 11 (9.73%), mourning *n* = 6 (5.31%), economic difficulties *n* = 5 (4.42%), intracouple conflicts *n* = 2 (1.76%), conflicts with family of origin *n* = 1 (0.88%), severe disease or accident of a family member *n* = 3 (2.65%), multiple stressful life events *n* = 24 (21.24%)] and in 50 women (62.5%) [job loss *n* = 5 (6.25%), mourning *n* = 7 (8.75%), economic difficulties *n* = 2 (2.5%), intracouple conflicts *n* = 4 (5%), severe disease or accident of a family member *n* = 4 (5%), multiple stressful life events *n* = 28 (35%)] in the PHQ-9 >4 group. Social support types were no support, *n* = 5 (4.4%), support by a partner, *n* = 6 (5.31%), support by relatives/friends, *n* = 18 (15.92%), and multiple support, *n* = 84 (74.33%) in the PHQ-9 ≤4 group, and no support, *n* = 2 (2.5%), support by partner, *n* = 5 (6.25%), support by relatives/friends, *n* = 28 (35%), and multiple support (i.e., by all of the above) *n* = 45 (56.25%) in the PHQ-9 > 4 group; overall, support type did not differ between the two groups, but there were more women with multiple support in the PHQ-9 ≤4 group at the expense of the support by relatives/friends type, which, however, merged into the multiple support type in this group.

Logistic regression of TEMPS-A affective temperaments identified the depressive temperament (OR: 1.310; *p*= 0.006) as a risk factor and hyperthymic temperament as a protective factor (OR: 0.871; *p* = 0.005) for depressive symptomatology during pregnancy. Results are shown in Table 2 and Table 3.

In our sample, there were eleven patients with current medical conditions (four hypertension, three diabetes, two fibromyalgia, and two generalised anxiety disorder); six were in the PHQ-9 ≤4 group and five in the PHQ-9 >4 group, with conditions about equally distributed. These patients were on antihypertensive medication (two nifedipine and labetalol and two methyldopa), pregabalin (four patients with fibromyalgia or generalised anxiety disorder), and oral hypoglycaemics (two metformin and one glibenclamide). Due to the low occurrence of significant medical conditions in our sample (5.7%), it is unlikely that drug intake could have affected results.

## 5. Discussion

Many studies show a significant presence of depressive symptomatology during pregnancy suggesting that monitoring mental health in this particular period should be considered a major public health issue. Antenatal depression has been found to constitute the strongest predictor of postnatal depression, which in turn predicts parenting stress [18]. Furthermore, maternal depression has a negative impact on physical, emotional, cognitive, behavioural, and social child development [19]. An Italian study found depressive symptomatology in 30.87% of women during the second trimester of pregnancy [20]. We here found more women in the PHQ-9 >4 group to belong to the first trimester, that could mean that the more severe group develops depressive symptoms early in pregnancy (or carries them over from the preceding period). A meta-analysis has indicated that depression has a prevalence of 25.6% in pregnant women during the COVID-19 pandemic [21]. A recent review reported that globally, perinatal mental health has worsened during the COVID-19 pandemic regardless of infection status [22].

In our sample, the prevalence of depressive symptomatology was 41.45% and that of depression was 9.85% (6.75% mild and 3.10% moderate depression). Our results are in line with findings by Biaggi et al. [1] but lower than the rate reported by Tomfohr-Madsen et al. [21]. A possible explanation might be the positive effect of periodic follow-up visits and decreased perception of pandemic, in that specific period, in Italy. Another explanation could be the resilience provided by expecting a child, which correlates with perceived stress and anxiety [23,24]. In fact, higher perceived stress and anxiety have been observed in prepandemic samples, compared to samples after the breakthrough of the pandemic [25,26,27], which we attributed to the increased ability of pregnant women to cope with novel generalised threats and mobilise inner resources, such as those related to resilience [26], probably based on increased brain plasticity, which is induced by pregnancy [27].

We have chosen a cut-off of >4 to identify mild depressive symptoms which may predict subsequent depression. Statistically significant differences between the two groups were found in gestational age, having an occupation, having a partner, past (lifetime) medical conditions, lifetime psychiatric disorders, family psychiatric history, stressful life events, and TEMPS-A mean scores. While having a job and a partner and being devoid of medical or psychiatric conditions and past stressful situations may intuitively boost resilience and increase the coping abilities of a person [28,29,30], thus reducing the severity of depressive symptoms [31], the different distribution of PHQ-9 ≤4 and PHQ-9 >4 in the three trimesters is more difficult to explain. It might be that women scoring >4 on the PHQ-9 carried their depressive symptoms over the period preceding pregnancy, or that they were more susceptible than women scoring ≤4 and developed depressive symptoms sooner.

In the symptomatic (PHQ-9 >4) group we found more stressful life events, particularly more multiple events, mourning, intracouple conflicts, and severe disease or accident of a family member. Job loss, economic difficulties, and conflicts with family of origin were present less often. Loss of job and economic difficulties may be addressed to a certain extent by social support [32,33], while conflicts with family of origin have a lesser impact compared with intracouple conflicts, although the two are interrelated [34].

Regarding the presence of support, we found no statistically significant differences between the PHQ-9 ≤4 and the PHQ-9 >4 groups (95.6% vs. 97.5%, respectively). With regard to the type of support, our findings showed similar rates in partner support, but more support from relatives/friends in the symptomatic (PHQ-9 >4) group (15.92% vs. 35%) and more multiple support in the PHQ-9 ≤4 group (74,33% vs. 56.25%). This might suggest that different types of support are not equivalent and it might be interesting to investigate it in a larger sample. Notably, with regard to the partner, there were statistically significant differences between the two groups (99.1% vs. 93.8%). This result is consistent with findings suggesting that a stable couple relationship is a protective factor from depression [35,36]. However, we should note that multiple support, which was preponderant in the no or minimal depressive symptoms (PHQ-9 ≤4) received all other support cases and this has determined the higher prevalence of support from relatives/friends in the more disadvantaged PHQ-9 >4 group.

Some studies have reported that depressive episodes are more frequent during the first and the third trimesters of pregnancy, probably as a consequence of stress of coping with becoming mothers and start a new life [1]. A review found that the prevalence of depression during the first trimester of pregnancy is similar to the prevalence in general population, but it is double during the second and third trimesters [37]. Another review showed that prenatal depression is more prevalent during the third trimester and less during the second one [38]. We found that the second trimester was the most frequent in both groups, but the first trimester was more represented in the symptomatic group for the reasons we hypothesised above.

Studies have shown that although many women with a past history of depression do not recur during the perinatal period, pregnancy is a time of life that should be carefully monitored [39,40]. In our sample, the different rate of psychiatric disorders lifetime between the two groups was statistically significant. Positive family psychiatric history was also more represented in the PHQ-9 >4 group.

Few studies have investigated the possible correlation of affective temperaments with mood disorders during the perinatal period. In our sample, mean scores on all affective temperaments but the hyperthymic were significantly lower in the less symptomatic PHQ-9 ≤4 group. Only depressive and hyperthymic temperaments were found to be a risk and a protective factor, respectively, for depressive symptomatology. Our results are consistent with a study that compared pregnant and non-pregnant women. One study has shown that women with bipolar disorder are at a higher risk of manic/mixed episodes during pregnancy, while their risk for developing a depressive episode is reduced [39]. Another study has shown that pregnancy favours hyperthymia and suggested that hyperthymic temperament might be protective for depressive symptomatology during this particular period of life [41]. In our groups, women screened negative for bipolar disorder. It could be hypothesized that different combinations of affective temperamental domains as well as psychosocial factors could represent vulnerability markers and help differentiating mood disorders. A study reported that hyperthymic temperament also seems to be a protective factor in the postpartum period [42]. Another one identified cyclothymic and anxious temperaments as possible risk factors for postpartum depression, independently from psychosocial factors [43]. In our sample, scores on both temperaments were lower in the PHQ-9 ≤4 group. Anxious, cyclothymic, depressive, and irritable temperaments were found to be related to more dysfunctional sleep patterns during the perinatal period [44]. Pregnancy seems to independently and negatively predict irritable, anxious, and cyclothymic temperament [45]; our results on the PHQ-9 ≤4 group are consistent with these findings.

## 6. Conclusions

This study confirmed a high prevalence of depressive symptomatology during pregnancy, indicating a complex aetiology. Mental health routine screenings during pregnancy should be implemented to identify vulnerable women and provide preventive and supportive interventions. It is fundamental to identify specific risk factors to develop adequate screening tools. Our results suggest that affective temperaments assessment could be a useful adjunctive instrument to predict depressive symptomatology during the peripartum period, although due to its length it cannot be recommended as a basic screen. A shorter version or a new, short questionnaire could be developed and included in basic screening. Affective temperament is a concept to incorporate in interviewing pregnant women with mood disturbances.

## Figures and Tables

**Figure 1 jpm-13-00211-f001:**
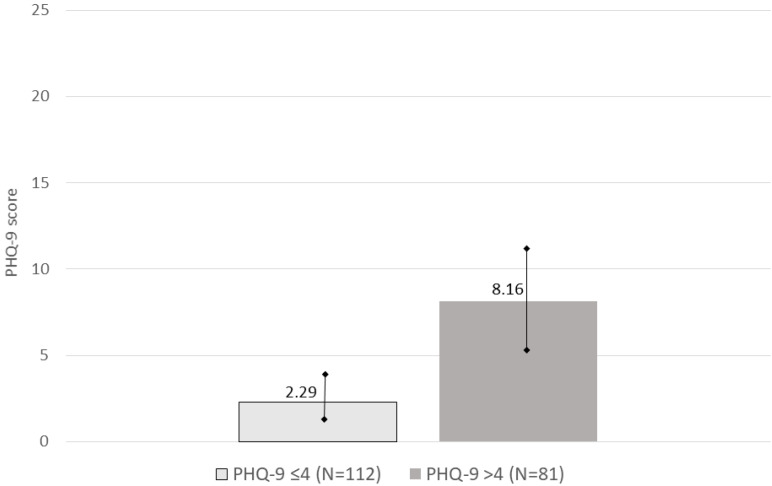
PHQ-9 scores. Groups subdivided according to the 4 cut-off. Bars represent standard deviation.

**Table 1 jpm-13-00211-t001:** Sociodemographic variables, epidemiologic characteristics, and temperament scores in the entire sample (*n* = 193).

Parameter	PHQ-9 ≤4 (*n* = 113)	PHQ-9 > 4 (*n* = 80)	*χ*^2^ or F	df	*p* Value
**Age**, yr (mean ± SD)	35.39 ± 5.50	33.77 ± 6.07	F = 3.58	1	0.060
**Gestational age**, trimester (*n* (%))					
I	9 (8%)	16 (20%)	*χ*^2^ = 6.32	2	**0.042**
II	79 (69.9%)	51 (63.7%)			
III	25 (22.1%)	13 (16.3%)			
**Educational level** (yr) (*n* (%))					
≤13	56 (49.6%)	38 (47.5%)	*χ*^2^ = 0.0794	1	0.778
>13	57 (50.4%)	42 (52.5%)			
**Occupation** (*n* (%))					
Yes	81 (71.7%)	45 (56.3%)	*χ*^2^ = 4.92	1	**0.027**
No	32 (28.3%)	35 (43.7%)			
**Partner** (*n* (%))					
Yes	112 (99.1%)	75 (93.8%)	*χ*^2^ = 4.48	1	**0.034**
No	1 (0.9%)	5 (6.2%)			
**Past pregnancies** (*n* (%))					
Yes	71 (62.8%)	50 (62.5%)	*χ*^2^ = 0.00221	1	0.963
No	42 (37.2%)	30 (37.5%)			
**Abortions** (*n* (%)) *					
Yes	34 (30.1%)	22 (27.5%)	*χ*^2^ = 0.152	1	0.696
No	79 (69.9%)	58 (72.5%)			
**Medical conditions (lifetime)** (*n* (%))					
Yes	27 (23.9%)	30 (37.5%)	*χ*^2^ = 4.17	1	**0.041**
No	86 (76.1%)	50 (62.5%)			
**Psychiatric disorders (lifetime)** (*n* (%))					
Yes	15 (13.3%)	27 (33.7%)	*χ*^2^ = 11.5	1	**<0.001**
No	98 (86.7%)	53 (66.3%)			
**Family psychiatric history** (*n* (%))					
Yes	32 (28.3%)	36 (45%)	*χ*^2^ = 5.71	1	**0.017**
No	81 (71.7%)	44 (55%)			
**Stressful life events** (*n* (%))					
Yes	52 (46%)	50 (62.5%)	*χ*^2^ =5.11	1	**0.024**
No	61 (54%)	30 (37.5%)			
**Support** (*n* (%))					
Yes	108 (95.6%)	78 (97.5%)	*χ*^2^ = 0.496	1	0.481
No	5 (4.4%)	2 (2.5%)			
**Psychometric assessment** ($\bar{x}$ ± SD)					
**TEMPS-A depressive**	4.89 ± 2.01	8.14 ± 4.13	F = 42.20	1	**<0.001**
**TEMPS-A cyclothymic**	2.42 ± 2.33	4.31 ± 3.71	F = 16.36	1	**<0.001**
**TEMPS-A hyperthymic**	11.10 ± 3.85	8.65 ± 4.31	F = 16.49	1	**<0.001**
**TEMPS-A irritable**	1.32 ± 1.86	3.24 ± 3.34	F = 21.66	1	**<0.001**
**TEMPS-A anxious**	4.04 ± 3.48	8.09 ± 5.03	F = 38.60	1	**<0.001**

Significant results in **bold** characters at last column. * 31 spontaneous, 3 planned in the PHQ-9 ≤4 group and 20 spontaneous and 2 planned in the PHQ-9 >4 group (*χ*^2^ = 0.0012; *p* = 0.973, not significant). *Abbreviations:* df, degrees of freedom; F, ANOVA’s coefficient; PHQ-9, Patient Health Questionnaire-9; SD, standard deviation; TEMPS-A, Temperament Evaluation of Memphis, Pisa, Paris and San Diego-Autoquestionnaire; $\bar{x}$, mean, *χ*^2^, chi-square.

**Table 2 jpm-13-00211-t002:** Logistic regression.

Parameter	Odds Ratio	95% C.I.	*p* Value
TEMPS-A depressive	1.310	1.079 to 1.591	**0.006**
TEMPS-A cyclothymic	1.068	0.889 to 1.284	0.481
TEMPS-A hyperthymic	0.871	0.790 to 0.960	**0.005**
TEMPS-A irritable	1.003	0.808 to 1.244	0.981
TEMPS-A anxious	1.095	0.959 to 1.251	0.179
Trimester	0.667	0.340 to 1.306	0.237
Occupation	0.573	0.258 to 1.274	0.172
Medical conditions	1.710	0.751 to 3.893	0.202
Psychiatric disorders	1.277	0.482 to 3.384	0.623
Family psychiatric history	1.098	0.480 to 2.513	0.824
Stressful life events	0.733	0.329 to 1.635	0.449
Partner	0.160	0.014 to 1.820	0.140

Significant results in **bold** characters. *Abbreviations*: TEMPS-A, Temperament Evaluation of Memphis, Pisa, Paris and San Diego-Autoquestionnaire, 95% C.I., 95% confidence intervals.

**Table 3 jpm-13-00211-t003:** Significant correlations between two TEMPS-A scales and the total score on the PHQ-9.

	TEMPS-A Depressive	TEMPS-A Hyperthymic	PHQ-9 Score
**TEMPS-A Depressive**	----		
**TEMPS-A Hyperthymic**	*r* = −0.257, *p* < 0.001	----	
**PHQ-9 score**	*r* = 0.598, *p* < 0.0001	*r* = −0.251, *p* < 0.001	-----

*Abbreviations:* TEMPS-A: Temperament Evaluation of Memphis, Pisa, Paris and San Diego-Autoquestionnaire; PHQ-9: Patient Health Questionnaire-9.

## Data Availability

The data presented in this study are available upon reasonable request to the corresponding author.

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
