# Peer review of "Depressive Symptoms during Pregnancy: Prevalence and Correlates with Affective Temperaments and Psychosocial Factors"

_jpm, 2023, doi:10.3390/jpm13020211_

Round 1
Reviewer 1 Report
- Comments shown in the attached file.

Author Response
Here we respond point-to-point, keeping your original observations. Please find changes in the text in red-coloured characters. Our responses are underneath each point raised and are in italics.
Manuscript ID: jpm-2123358 Title: Depressive symptoms during pregnancy: prevalence and correlates with affective temperaments and psychosocial factors
The manuscript addresses a relevant topic from a clinical and social point of view, deserving publication after improvement.
We thank Reviewer for appreciation.
The synthetic style of writing left out precious information about the method (selection of participants, context of their evaluation, systematization and data analysis strategies). There was no information about the delineation of the groups, named as control and clinical group. Does the type of methodological design used allow this type of denomination in the groupings of the women evaluated? It will be necessary to justify this technical decision.
We thank Reviewer for this observation and added the required information in the appropriate section. We actually never mentioned a clinical group, while we inappropriately referred to the PHQ-9 ≤4 as the “control” group. We corrected. The subdivision according to the 4 cutoff is legitimate, according to the PHQ-9, which is a validated instrument. It was already specified in the text that groups were subdivided according to the PHQ-9 in the Statistical Methods, but we repeated this above for clarity.
The results are presented in a descriptive and analytical way, considering the objectives, but were not sufficiently interpreted. The authors limited themselves to presenting tables and repeating data in the text, without offering their interpretative analysis of these findings. This requires extensive revision of the manuscript.
We tried to avoid overlapping findings and rewrote completely the Results section (adding surreptitiously some interpretation, to suit your suggestion). We hope the section is more readable now.
The discussion tended to repeat the results, with little debate in relation to the information present in the scientific literature of the área.
We thank Reviewer for this suggestion. We complied.
It will be necessary to improve this process and to review specific aspects such as:
- Line 70 - reference cited should be to (8) and not (7).
We thank Reviewer for this observation. We corrected the error.
- Line 74 - reference cited should be (7) and not (8).
As above.
- Line 78 - it will be necessary to justify the argumentation presented.
We thank Reviewer for this suggestion. We modified the sentence to stress our point.
- Figure 1: Reducing decimals in the mean value of data (for clarity of information)
We thank Reviewer, we reduced decimals to two.
- Paragraphs 151-184 repeat data from Table 1. The text should be replaced and only describe the essentials, without replicating information already present in the table. It will be better to interpret the data than just describe them, highlighting the factors associated with maternal depression (to support the regression analysis performed).
We thank Reviewer for this suggestion, which we endorse. We replaced most of the repeated text, following your suggestion.
- lines 222-223 - explain further - data is confusing or the text is unclear.
We agree with Reviewer, we revised extensively.
Review, please!
We thank Reviewer for thoughtful suggestions. We hope that you will find the revised version to be suitable for publication.
Reviewer 2 Report
Provide more specifics on recruitment - how many declined, for example.
What constitutes stressful life events? What about medical conditions?
Is the TEMPS-A something that should be incorporated in screening all pregnant women?
What were diagnoses of patients? what meds were they on?
Author Response
Here we respond point-to-point, keeping your original observations. Please find changes in the text in red-coloured characters. Our responses are underneath each point raised and are in italics.
Comments and Suggestions for Authors
Provide more specifics on recruitment - how many declined, for example.
We thank Reviewer for this suggestion. We added how many declined and how many were excluded based on other exclusion criteria.
What constitutes stressful life events? What about medical conditions?
We specified which was the publication on which we based our concept of stressful life events. In Paykel’s list, serious medical conditions are considered major life stresses. In page 4, lines 140-145 we report “Stressful life events were present in 52 (46%) of women in the PHQ-9 ≤4 group [job loss n=11 (9.73%), mourning n=6 (5.31%), economic difficulties n=5 (4.42%), intracouple conflicts n=2 (1.76%), conflicts with family of origin n=1 (0.88%), severe disease or accident of a family member n=3 (2.65%), multiple stressful life events n=24 (21.24%)] and in 50 women (62.5%) [job loss n=5 (6.25%), mourning n=7 (8.75%), economic difficulties n=2 (2.5%), intracouple conflicts n=4 (5%), severe disease or accident of a family member n=4 (5%), multiple stressful life events n=28 (35%)]”. In our patients there were no psychoses or major medical conditions, despite the fact that the latter was not an exclusionary criterion.
Is the TEMPS-A something that should be incorporated in screening all pregnant women?
We don’t believe that it should be included in all screenings, but it could help understanding them better. However, pregnant women are not so eager to complete 110-item questionnaires, so to extend this to every pregnant woman would be hard to get through. We added some text to underline this point. Thank you for noticing.
What were diagnoses of patients? what meds were they on?
There were eleven patients with medical conditions taking appropriate drugs for their conditions. We specified this in the text. We added conditions and medications, they could not have affected results, as they were equally distributed. We thank Reviewer for constructive comments. We hope you will consider our revised version as meeting criteria for publication.
Round 2
Reviewer 1 Report
This version of the manuscript is improved, with important detail on the methodological steps taken by the researchers in their analysis.
The presentation and discussion of the data are clearer and achieved relevant debate with the scientific literature in the area. There are few aspects of the text in need of minimal revision, pointed out in the comments of the text sent at this time.

Author Response
Dear Reviewer,
We here respond to your comments in italics, just underneath each point raised, which we include in this response. Please find our changes in red characters in the new version (we turned black the revised version and highlighted changes in red).
This version of the manuscript is improved, with important detail on the methodological steps taken by the researchers in their analysis.
We thank you for your positive impression.
The presentation and discussion of the data are clearer and achieved relevant debate with the scientific literature in the area. There are few aspects of the text in need of minimal revision, pointed out in the comments of the text sent at this time.
We thank you and revised our text carefully. We hope you will like the new version.
Reviewer 2 Report
there are 112 patients in the PHQ<4 sample but there are results for 113 patients.
there are 81 patients in the PHQ>4 sample but results for 80 were recorded, according to the table
were abortions spontaneous or planned?
Stressful life events needs to be better delineated. All of us have stress in our lives. Many couples have conflicts, or conflicts with family. What does economic difficulty mean?
what were lifetime medical conditions?
line 185 A meta-analysis has indicated that depression has a prevalence of 25.6% - in the population, in pregnant women, or in some other cohort?
What would be next steps? if the TEMPS-A is not a good tool in practice, what would be? Otherwise the paper seems to state that there should be universal depression screening in the care of pregnant women, which is well known as your paper cited several references.
Author Response
Dear Reviewer 2,
We thank you for your request for further revision, as this will help settling some issues. I here respond to your queries by keeping them in this response and responding to each of them just underneath the point raised. This is to facilitate you in reviewing the new version. We put the already revised version in black-coloured characters and then highlighted the new changes in red characters. Our responses are in italics, while changes in the text appear in red. We hope you will find the new version acceptable for publication.
there are 112 patients in the PHQ<4 sample but there are results for 113 patients.
There were 113 women in the at least 4 sample. We corrected. Results in the Table were correctly adding to 113 for all categorical variables.
there are 81 patients in the PHQ>4 sample but results for 80 were recorded, according to the table
There were 80 women in the PHQ-9>4 sample. We corrected. Results in Table 1 were correctly adding to 80 for all categorical variables. We also corrected in the text, these were typos. Percentages were correct, you may control if you wish.
were abortions spontaneous or planned?
Most were spontaneous, few were planned. We specified these in the Table's footnotes. There were no significant intergroup differences for the type of abortion variable.
Stressful life events needs to be better delineated. All of us have stress in our lives. Many couples have conflicts, or conflicts with family. What does economic difficulty mean?
We further specified in the text. Economic difficulty is hardship, not having enough money to meet needs.
what were lifetime medical conditions?
Major surgery (abdominal or thoracic) and severe infections (severe varicella with sequelae, meningitis, and tuberculosis). We did not add this in the text to avoid confusion.
line 185 A meta-analysis has indicated that depression has a prevalence of 25.6% - in the population, in pregnant women, or in some other cohort?
In pregnant women during these times. You are right, our phrasing could have been confusing, we specified in the text.
What would be next steps? if the TEMPS-A is not a good tool in practice, what would be? Otherwise the paper seems to state that there should be universal depression screening in the care of pregnant women, which is well known as your paper cited several references.
We did not say it's no good. On the contrary, we stress that affective temperament should be taken into account in visiting a pregnant woman with mood issues. We only state that, as is, the TEMPS-A is difficult to handle with pregnant women, as they all have much to think about and don't wish to pass their time filling-in long questionnaires. The next step could be to provide a shorter affective temperament questionnaire that women would like to fill-in. A shorter version of the TEMPS exists, but about 40 questions make it a long questionnaire to get through. Furthermore, its developer died and I don't see who can validate a shorter version. We added some text to address this issue.
Thank you again for your efforts.